# Risk Assessment of Effects of Essential Oils on Honey Bees (*Apis mellifera* L.)

**DOI:** 10.3390/insects16030303

**Published:** 2025-03-14

**Authors:** Joel Caren, Yu-Cheng Zhu, Quentin D. Read, Yuzhe Du

**Affiliations:** 1Jamie Whitten Delta States Research Center, United States Department of Agriculture, Agricultural Research Service, Stoneville, MS 38776, USA; joel.caren@usda.gov (J.C.); yuzhe.du@usda.gov (Y.D.); 2United States Department of Agriculture, Agricultural Research Service, Southeast Area, Raleigh, NC 27606, USA; quentin.read@usda.gov

**Keywords:** essential oil, honey bee, enzymes, insecticide, biopesticide, toxicity

## Abstract

The impact of synthetic pesticides on non-target organisms has led to increased interest in environmentally friendly pest control methods, such as essential oils. To ensure these natural compounds are safe for pollinators, it is essential to assess their effects on honey bees. This study assessed the impacts of one commercially available essential oil mixture (EcoTec+) and four essential oil components (β-bisabolene, cinnamaldehyde, 1,8-cineole, and eugenol) on honey bee workers in a lab. EcoTec+ increased protective protein. Bisabolene decreased protective enzymes with no lethal effect. Cinnamaldehyde had mixed effects on protective enzyme activity and displayed toxicity when ingested. Cineole also had mixed effects on two of the four enzymes measured. Eugenol suppressed two protective enzymes and was toxic on contact. None of the compounds interacted significantly with the tested pesticides or inhibitors. Collectively, the essential oils studied did not pose a significant danger to honey bees and seem to be a generally safe alternative to certain synthetic pesticides.

## 1. Introduction

The imperative to feed and maintain the health of an increasing world population has spurred the growth of the pesticide industry. Synthetic pesticides play a vital role in safe-guarding crops from pests both pre- and post-harvest as well as in protecting humans from disease-carrying insects. However, the extensive use of specific synthetic pesticides leads to pests developing resistance [1] through a variety of mechanisms [2], necessitating a constant search for new pesticides and pest control methods [3]. Additionally, many synthetic pesticides are acutely toxic to non-target organisms such as honey bees [4] and pose risks to humans who handle them [5]. There has also been a growing push for more environmentally friendly methods of controlling agricultural pests [6,7].

This push has intensified interest in essential oils (EOs) as potential biopesticides. EOs are typically mixtures of insoluble secondary metabolites extracted from plants, primarily consisting of terpenes, terpenoids, and aromatic compounds. Recognized for centuries for their various beneficial properties [8], including insecticidal and repellent effects, EOs are considered an effective and safe alternative to current synthetic pesticides [9]. Moreover, they can also be sustainably extracted from agricultural waste [10].

Hence, research into the efficacy of essential oils has surged in recent years [11], with studies evaluating their effectiveness against a wide variety of pests including bed bug [12], mosquito [13,14,15,16,17,18], granary weevil [19,20], *Triatoma infestans* [21], poultry red mite [22], spider mite [9,23,24], silverleaf whitefly [9], aphid [9,23,25,26], house fly [23,27], cotton leafworm [23,26,28], fire ant [29,30], scabies mite [31], cotton, thrips [32], mealworm [26], tick [33,34,35], leafhopper [36], red flour beetle [37,38], cigarette beetle, booklouse [38], and nematode [39]. Additionally, some EOs have shown significant anti-fungal properties [40,41].

To utilize these potential biopesticides effectively, it is crucial to evaluate their risks to non-target organisms [42], particularly *Apis mellifera*, the Western honey bee. This assessment is particularly important as EOs are considered for controlling Varroa mites, and several studies have been conducted to determine the usefulness of EOs as acaricides. For example, Khajehali et al. [43] reported that EO treatments for mites were comparable in efficacy to synthetic chemicals. Narciso et al. [44] found no difference in colony health between EO-treated hives and controls. Bava et al. [45] noted moderate acaricidal efficacy but “not sufficient for the commercialization of a product”. However, Alsaadi et al. [46] identified several effective EO treatments for Varroa, and other researchers have found that some EOs can be repellent, even if they do not kill the mite [47,48]. Thus, while researchers have begun to assess the risks [49,50] and benefits [51,52] of EOs for honey bees, further direct research on EO biopesticide effects is needed [53].

Because most insecticides are applied by foliar sprays, honey bee foragers may be exposed directly to field sprays, and other resident young bees may be indirectly adversely impacted through the ingestion of contaminated pollens and nectars. In this study, we evaluated one commercially available EO mixture marketed as a broad-spectrum pesticide (EcoTec+) and four EO component compounds identified as possible active ingredients (a.i.s) (β-bisabolene, cinnamaldehyde, 1,8-cineole, and eugenol for crop pest control) using both spraying (mainly contact toxicity) and feeding (stomach poisoning) methods to treat honey bee workers in a lab setting to assess the acute toxicity (mortality) of the EOs. We also assayed combinations of each compound with enzyme inhibitors and representative synthetic pesticide formulations. Sublethal effects were also examined by measuring several biochemical (enzyme activities) and molecular (gene expressions) responses of the honey bee to the chemicals.

## 2. Materials and Methods

### 2.1. Honey Bees

Italian honey bees (*Apis mellifera* ligustica), mostly purchased as packages over the past several years from Bemis Honey Bee Farm (Little Rock, AR, USA), were kept in two locations: one at the edge of an agricultural research field seven miles south of Stoneville, MS, USA, and the other in a wildlife management area three miles north of Stoneville. The bottom board of each hive has an oil trap (35 × 45 cm tray filled with vegetable oil) below the metal screen for monitoring and controlling Varroa mite (*Varroa destructor* A.T.) and small hive beetle (*Aethina tumida*). To obtain approximately 12,500 bees for bioassays, deep frames with over 50% coverage of healthy sealed brood were collected from several strong colonies and transferred to a lidded container vented by large cutouts covered by mesh screen. The container was kept in a laboratory incubator at 33 ± 0.5 °C and 60% ± 3 RH in the dark. Newly emerged worker bees from different colonies were pooled and transferred daily in groups of 20 to cages fashioned from 500 mL round wide-mouth polypropylene jars (D × H: 9.3 × 10 cm) with a 3 × 8 cm section of plastic comb foundation attached vertically to the inside of the cage. The caged bees were provided with water and 50% (*w*/*v*) sucrose solution in two 20 mL scintillation vials with perforated lids placed upside down on the top of each cage. These caged bees were maintained in incubators under the same conditions described for sealed brood frames.

### 2.2. Chemicals

Formulated insecticides imidacloprid (Advise^®^ 2FL, Winfield Solutions, St. Paul, MN, USA), acephate (Bracket^®^97, Winfield Solutions, St. Paul, MN, USA), bifenthrin (Brigade^®^ 2EC, FMC Corporation, Philadelphia, PA, USA), and oxamyl (Vydate^®^, Corteva, Indianapolis, IN, USA) were purchased from local agricultural chemical suppliers and stored in a refrigerator (6 ± 1 °C). These formulations represent four insecticide classes (neonicotinoids, organophosphates, pyrethroids, and carbamates, respectively) commonly recommended for major cotton insect control [54]. These representative insecticides were used for tests of interaction with the essential oils being evaluated.

Essential oil components β-bisabolene (18.27% in *Alpinia galanga*), cinnamaldehyde (64.66% in *Cinnamomum verum*), 1,8-cineole (eucalyptol, 10.17% in *Alpinia galanga* [13]), and eugenol (18.69% in *Cinnamomum verum* [55]) were purchased from Grainger (Lake Forest, IL, USA) and Thermo-Fisher Scientific (Waltham, MA, USA) and stored at 6 ± 1 °C. EcoTec+, a mixture of essential oils labeled for pest control, was obtained from Forestry Distributing (Longmont, CO, USA) and stored at the same conditions. Essential oil compounds are insoluble, so dimethyl sulfoxide (DMSO) was purchased from Thermo-Fisher Scientific and was used at 2% to facilitate solubility.

### 2.3. Bioassays

Cages with more than three dead bees were not used for experimentation. Just before a bioassay was started, dead bees were counted and excluded from the total number of bees tested in that replicate. Four types of bioassays were conducted for each chemical: contact gradient, feeding gradient, inhibition (contact), and mixtures (contact). Bees were sampled from contact bioassays at 24 h or from feeding bioassays at 7 days and immediately frozen at −80 °C for enzyme activity and qRT-PCR assays. Because of a lack of directly comparable data for honey bees, the LD_50_ value for mosquitos [13] was selected as a standard concentration for comparison of EO active ingredients, corresponding to 3.1 mg/mL (EcoTec+), 5.8 mg/mL (bisabolene), 8.54 mg/mL (cinnamaldehyde), 3.25 mg/mL (cineole), and 5.8 mg/mL (eugenol).

To measure contact toxicity, a modified Potter Spray Tower was used to deliver 500 μL of the pesticide or essential oil solution or 2% DMSO in dH_2_O to each cage containing 20 bees. The sprayer was set at 10 psi with spray distance of 22 cm, ensuring uniform deposition of mist on bees and the inner surface of the container [4]. After spraying, caged bees were maintained in an incubator as described above. Mortality was recorded 24 h and 48 h after treatment, and surviving bees were sampled at 24 h.

To study possible interactions with other chemicals, each EO was mixed with a series of three inhibitors, namely piperonyl butoxide (PBO), triphenyl phosphate (TPP), and diethyl maleate (DEM), as well as four synthetic pesticides representing different classes of action: Advise Four (neonicotinoid), Bracket97 (organophosphate), Brigade 2 EC (pyrethroid), and Vydate (carbamate). Inhibitors were at 1% *w*/*v*, and pesticides were at LC_20_ (137.22 mg/L, 90.9 mg/L, 130.65 mg/L, and 161.56 mg/L, respectively) [4]. Essential oil compounds were also used at LC_20_ or at referred concentrations in mixtures.

To measure feeding toxicity, 50% sucrose vials were replaced with vials containing 50% sucrose, 2% DMSO, and the compound of interest. Controls contained 50% sucrose and 2% DMSO in dH_2_O without added chemicals. After replacing the vials, bees were maintained as described, and mortality was recorded at 24 h, 48 h, and 7 days. Surviving bees were sampled at 7 days. Sucrose vials were weighed at the beginning and end of the assay to determine consumption over 7 days.

### 2.4. Enzyme Assays

Bees sampled from bioassays and subsequently frozen were assayed for esterase (EST), glutathione S-transferase (GST), acetylcholine esterase (AChE), and soluble protein following the procedures outlined previously [56]. Specifically, clarified homogenate was extracted by homogenizing three bees minus the abdomens per sample in 1 mL 0.1 M sodium phosphate pH 7.2 with Halt protease inhibitor (cat # 1861279, Thermo Scientific, Waltham, MA, USA) and 0.3% Triton X-100 (cat # X100-100ml, Sigma Aldrich, St. Louis, MO, USA), then centrifuging for 15 min at 14,000 rpm and 4 °C. The supernatant was transferred to a new tube and centrifuged again at the same specifications. This supernatant was then diluted by 1:3 with the above extraction buffer.

To measure EST activity, the clarified homogenate was diluted by 1:4 in extraction buffer, then 15 µL was added to an assay plate (cat # 9017, Corning, Glendale, AZ, USA), followed by 135 µL 0.3 mM α-naphthyl acetate (Sigma-Aldrich, cat # N8505-5G), and then incubated 30 min at 37 °C. The reaction was stopped by adding 50 µL Fast Blue B salt (cat # D9805-10g, Sigma-Aldrich) and then incubating for 15 min at room temperature. Absorbance was read at 570 nm.

To measure GST, 80 µL potassium phosphate pH 6.5 + 1 mM EDTA was added to a Corning assay plate, and then 10 µL clarified homogenate, 100 µL 20 mM L-glutathione, and 10 µL 40 mM 1-chloro-2,4-dinitrobenzene were added and the mixture was vortexed at a low speed for 60 s. Absorbance was read at 340 nm for 10 min with a 20 s interval, and then the change in absorbance over time was recorded.

To measure AChE, 15 µL clarified homogenate was added to a Corning assay plate. Then, acetylthiocholine iodide was added to a final concentration of 0.25 mM and 5,5′-dithio-bid-2-nitrobenzoic acid was added to a final concentration of 0.4 mM. The mixture was vortexed at a low speed for 60 s. Absorbance was measured at 412 nm for 15 min with a 20 s interval, and the change in absorbance over time was recorded.

### 2.5. qRT-PCR of Detoxification Genes

To measure representative P450 transcript levels, one bee abdomen per sample (three samples per treatment) was homogenized using Benchmark Scientific (Sayreville, NJ, USA) pre-filled tube kits containing sterile, nuclease-free 3.0 mm zircon beads (cat # D1032-30) in a Fisherbrand Bead Mill 24 (cat # 15-340-163, Fisher Scientific, Waltham, MA, USA), and total RNA was extracted using Bio-Rad (Hercules, CA, USA) Aurum™ Total RNA mini-kits (cat # 732-6820). RNA was immediately reverse-transcribed using Verso cDNA Synthesis kits (cat # AB-1453/B, Thermo Scientific), and qRT-PCR was run in an Applied Biosystems (Waltham, MA, USA) QuantStudio 3 using PowerUp™ SYBR™ Green Master Mix (cat # A25742, Applied Biosystems, Waltham, MA, USA). Four P450 genes, selected to represent the important detoxification activity of the CYP families [57,58], were analyzed: CYP-6A13, CYP-6AQ1, CYP-9Q1, and CYP-9Q2 (see Table 1). Ribosomal protein 49 (RP-49) was used as housekeeping reference. Data were processed using the 2^−∆∆Ct^ method.

### 2.6. Statistical Analyses

Data were processed using Microsoft Excel. LC_50_ value was calculated by Probit using the BioRssay library in R, version 1.1.0 [59], which transforms raw data with the Abbott correction [60] if control mortality is high. Linear, log-linear, and exponential decay models were fitted to each response variable as a function of dose. For the log-linear models, negative controls were adjusted by adding 0.01 before log transformation of the independent variable. Likelihood ratio tests were conducted to compare each fitted model to a null model. Statistical analyses were performed using R version 4.3.0. Standard error was calculated by finding the standard deviation of each mean and dividing by the square root of the sample size.

## 3. Results

### 3.1. Effects of EcoTec+ on Honey Bees

Contact (spray treatment) bioassays of EcoTec+ resulted in no significant mortality, even at 8× the recommended field concentration (3.1 mL/L), which was the highest concentration tested. Similarly, feeding bioassays showed minimal mortality at an 8× field concentration. No significant interactions were observed between EcoTec+ and the tested inhibitors (piperonyl butoxide, triphenyl phosphate, and diethyl maleate) nor between EcoTec+ and representative synthetic pesticides. The enzyme activity levels for EST, GST, and AChE in the treated groups were comparable to the controls. However, when the mixture was fed to bees, the CYP-6A13 P450 transcript levels increased linearly with an increasing EcoTec+ concentration at the rate of 0.059 per 1 mg/mL of increase in the chemical (F_1,16_ = 25.09, *p* < 0.001, R^2^ = 0.61) (Figure 1). All non-significant statistical data (Df, F value, *p* value, and R^2^) are shown in Appendix A. All non-significant bioassay mortality data are listed in Appendix A.

### 3.2. Effects of Bisabolene on Honey Bees

Both contact and feeding of bisabolene at 8× LD_50_ for mosquitos (5.8 mg/mL) [13] resulted in mortality within one standard deviation of the control levels, and no significant interaction was detected between bisabolene and inhibitors or the representative pesticides. When bees were fed bisabolene, the EST levels decreased at a rate of 0.371 nmol/min/mg per doubling of bisabolene concentration (F_1,16_ = 27.22, *p* < 0.001, R^2^ = 0.63) (Figure 2a). Similarly, when bees were sprayed with bisabolene, AChE decreased at the rate of 0.064 nmol/min/g with each doubling of the bisabolene concentration (F_1,19_ = 22.00, *p* < 0.001, R^2^ = 0.52) (Figure 2b). GST was unaffected by feeding, but contact increased GST activity by 0.492 nmol/min/mg for each mg/mL of increase in the compound (F_1,19_ = 8.02, *p* < 0.05), but the correlation was low (R^2^ = 0.297) (Figure 3). Contact caused the P450 CYP-6A13 transcript levels to decrease by 0.309 for each doubling of the bisabolene concentration (F_1,19_ = 18.05, *p* < 0.001), though the correlation was low (R^2^ = 0.487), (Figure 4). However, contact caused the P450 CYP-9Q1 transcript levels to increase by 0.243 per mg/mL of increase in the chemical (F_1,19_ = 34.66, *p* < 0.001, R^2^ = 0.646), (Figure 5). All non-significant statistical data (Df, F value, *p* value, and R^2^) are shown in Appendix A. All non-significant bioassay mortality data are listed in Appendix A.

### 3.3. Effects of Cinnamaldehyde on Honey Bees

Contact exposure to cinnamaldehyde did not result in significant mortality at 8× LD_50_ for mosquitos (8.54 mg/mL) [13], while feeding bioassays showed mortality at LC_50_ = 9.6 mg/mL (Table 2), within the US-EPA category of Practically Nontoxic (LC_50_ > 5.0 g/L) [61]. No significant interaction was observed between cinnamaldehyde and inhibitors or the representative pesticides. Cinnamaldehyde did not significantly affect the enzyme activity of EST or GST on either contact or feeding bioassays. Contact led to a linear increase in AChE activity at the rate of 0.0024 nmol/min/g per mg/mL of increase in the chemical (F_1,19_ = 30.67, *p* < 0.001, R^2^ = 0.617) (Figure 6a). However, feeding resulted in a decrease in AChE activity at the rate of 0.096 nmol/min/g with each doubling of the cinnamaldehyde concentration (F_1,16_ = 37.15, *p* < 0.001, R^2^ = 0.699) (Figure 6b). The transcript levels of the P450 gene CYP-9Q1 decreased by 0.324 with each doubling of the compound concentration (F_1,16_ = 11.14, *p* < 0.01), though the correlation was low (R^2^ = 0.41) when bees were fed the compound (Figure 7). No significant effects were observed for other tested P450 genes. All non-significant statistical data (Df, F value, *p* value, and R^2^) are shown in Appendix A. All non-significant bioassay mortality data are listed in Appendix A.

### 3.4. Effects of Cineole on Honey Bees

Exposure to cineole did not cause significant mortality in honey bee workers at 8× LD_50_ for mosquitos (3.25 mg/mL) [13] whether through contact or feeding. No apparent interactions were observed between cineole and the tested inhibitors or pesticides. Also, cineole did not significantly affect GST or AChE activity. However, feeding caused a linear decrease in esterase at the rate of 0.051 nmol/min/mg for each mg/mL of increase in the compound (F_1,16_ = 5.40, *p* < 0.05), but the correlation was low (R^2^ = 0.252) (Figure 8). Contact with cineole resulted in a 0.088 decrease in the transcript levels of CYP-6AQ1 with each doubling of the cineole concentration (F_1,28_ = 26.49, *p* < 0.001), though the correlation was low (R^2^ = 0.45) (Figure 9). Similarly, contact exposure led to a decrease in the CYP-9Q1 transcript levels of 1.67 with each doubling of the cineole concentration (F_1,19_ = 5.09, *p* < 0.05), with a low correlation (R^2^ = 0.21) (Figure 10a). However, feeding cineole caused CYP-9Q1 to increase linearly at the rate of 0.232 per mg/mL increase in the compound (F_1,16_ = 92.98, *p* < 0.001, R^2^ = 0.853) (Figure 10b). Similarly, CYP-9Q2 increased linearly at a rate of 0.080 per mg/mL of increase in cineole (F_1,16_ = 36.81, *p* < 0.001, R^2^ = 0.568) (Figure 11). All non-significant statistical data (Df, F value, *p* value, and R^2^) are shown in Appendix A. All non-significant bioassay mortality data are listed in Appendix A.

### 3.5. Effects of Eugenol on Honey Bees

Contact bioassays of eugenol resulted in honey bee mortality with LC_50_ = 86 mg/mL, corresponding to LD*_50_* = 135 µg/bee [4], within the US-EPA category of Practically Nontoxic (see Table 3). However, feeding did not cause significant mortality at the highest concentration tested relative to the control. No interaction between eugenol and inhibitors or representative pesticides was observed, and GST and P450 transcript levels were not significantly affected by the compound. However, contact with eugenol caused a linear decrease in esterase at the rate of 0.0438 nmol/min/mg for each mg/mL of increase in the compound (F_1,19_ = 40.6, *p* < 0.001, R^2^ = 0.681) (Figure 12). In addition, contact with eugenol caused a linear decrease in AChE at the rate of 0.0032 nmol/min/g for each mg/mL of increase in the compound (F_1,19_ = 8.72, *p* < 0.01), but the correlation was low (R^2^ = 0.315) (Figure 13). All non-significant statistical data (Df, F value, *p* value, and R^2^) are shown in Appendix A. All non-significant bioassay mortality data are listed in Appendix A.

## 4. Discussion

We tested one commercially available EO pesticide and four EO component compounds with reported insecticidal properties against honey bee workers (*A. mellifera*) as a representative nontarget organism. Mixtures of pesticide chemicals and inhibitors are commonly used for synergistic activity, so we also tested mixtures of these five compounds with three inhibitors and pesticides representing four of the most common classes of insecticides. Overall, the EOs studied here demonstrated no significant acute lethal toxicity to honey bees. Cinnamaldehyde showed stomach (oral) toxicity when fed to honey bees, and eugenol was toxic on contact (spray), but these toxic effects were only seen with high concentrations of the compounds (LC_50_ = 9.6 mg/mL and 86 mg/mL, respectively). EcoTec+, cineole, and bisabolene did not induce significant mortality at the highest concentrations tested. Biochemical effects varied among the compounds but remained largely sublethal.

EcoTec+ is labeled as a “broad-spectrum insecticide and miticide”, containing rosemary oil (10%), geraniol (5%), a monoterpenoid, and peppermint oil (2%). It has shown “extensive university and grower field-testing proven efficacy” and is “as effective as conventional pesticides” according to information from the Hydrobuilder website (https://hydrobuilder.com/brandt-ecotec-plus-organic-insecticide-miticide-2-5-gallons.html, accessed 18 December 2023). EcoTec+ was not specifically tested against target organisms in our study, so the validity of these claims cannot be confirmed here. Among the components of EcoTec+, rosemary oil’s primary component is cineole (29%) [62], which was not toxic to bees at the concentrations we studied. Geraniol toxicity to honey bees was measured at LD_50_ (contact) = 290 µg/bee and LD_50_ (feeding) = 43 µg/bee by Santos [63]. When bees were sprayed with EcoTec+ at 24.8 g/L, each bee was exposed to 2.0 µg of geraniol, well below the LD_50_ value. At an average feeding rate of 22 µL/bee/day, each bee would have ingested 191 µg of geraniol over the course of 7 days, yet mortality was not significantly different from the control. Peppermint oil, which is primarily made up of menthol (44%) and p-menthone (29%), is toxic to honey bees at LC_50_ (contact) = 11,900 ppm and LC_50_ (feeding) = 5471 ppm [64]. At the highest concentration we tested, bees were exposed to only 496 ppm peppermint oil, well below the LC_50_ value for either contact or feeding. Overall, the laboratory bioassays conducted showed no measurable acute toxicity to honey bees, though CYP-6A13 P450 transcripts increased when bees were fed the compound. This increase indicates that CYP-6A13 likely plays a role in the processing of one or more of the constituent EOs of EcoTec+.

Bisabolene, a sesquiterpenoid, has shown insecticidal activity against mosquito larvae, with an LC_50_ value below 5 µg/mL [65]. However, da Silva et al. [49] reported that ginger oil, containing 10% bisabolene, had an LC_50_ value of 22.0% for honey bees in contact bioassays. Given a density of 0.871 g/mL, this corresponds to 192 mg/mL ginger oil, or 19.2 mg/mL bisabolene, yet our own tests with up to 46.4 mg/mL bisabolene caused no significant mortality. We only saw sublethal effects on EST, GST, AChE, and P450. This difference may be due to our use of the isolated compound compared to da Silva’s use of a whole essential oil.

Cinnamaldehyde, a flavonoid, was reported to be potent against nematodes at a lower concentration (0.8 mg/mL caused 100% mortality) than Cypermethrin (2 mg/mL caused 100% mortality) [39], which is over 10× lower than the LC_50_ value observed for honey bees in our study (9.6 mg/mL). Interestingly, cinnamaldehyde up-regulated GST transcripts but not AChE in nematodes [39], while we observed significant effects on AChE but not GST in honey bees. Surprisingly, the effects on AChE were mixed: contact caused AChE to increase, while feeding caused it to decrease. The suppression observed when bees were fed is likely the result of direct inhibition, but studies have shown that AChE is a stress indicator in honey bees [66]. Therefore, increased AChE activity may be a stress response to contact with the irritating chemical. Previous studies showed no elevated mortality in honey bee at 1.25 mg/mL feeding [67]. Cinnamaldehyde also suppressed CYP-9Q1.

Cineole (eucalyptol), a monoterpene, has previously been shown to be “relatively innocuous” when fed to honey bee workers [68], which is in agreement with our own data. However, it has demonstrated synergy with the natural alkaloids matrine and oxymatrine against *Plutella xylostella* and *Tetranychus urticae* [25]. Other studies have reported synergism with pyrethroids against mosquito [14,22]. However, those studies used whole essential oil plant extracts, and we used purified active ingredients in our experiments. Additionally, cineole has shown synergy with thymol in controlling dog ticks [35]; a future investigation might explore the effects of such combinations on honey bees. We did not detect any synergy between cineole and other tested xenochemicals, including the pyrethroid Brigade 2EC. Cineole has also been reported to inhibit AChE in a similar manner to organophosphates and carbamates in mosquitos [69]. Similarly, cineole had an anti-acetylcholinesterase activity of IC_50_ = 32.8 µg/mL in bacteria [70] and 0.67 mM in human erythrocytes [71]. Our experiments did not reveal the consistent inhibition of AChE in honey bees, though we found that it decreased esterase activity and suppressed CYP-6AQ1. The CYP-9Q1 response, however, was mixed: contact suppressed the transcript, but feeding caused a linear increase. The CYP-9Q2 transcript was also increased with an increasing compound concentration. Thus, CYP-9Q1 and CYP-9Q2 both likely play a part in processing cineole.

Eugenol, a phenylpropanoid, has been found to hyperactivate *Triatoma infestans* nymphs, potentially acting as a ligand of octopamine receptors [22]. A study of eugenol’s effect on the American cockroach agreed on this mode of action [72], while a docking study in mosquitos found that eugenol’s docking energy with octopamine receptor and AChE is comparable to that of their native substrates [73]. In honey bees, it was shown to significantly reduce Nosema infection without harming the colony in one study [74], though other lab tests revealed significantly high honey bee mortality compared to the control when the compound was fed [67]. Our own bioassays did not replicate those results. Rather, we saw more significant mortality from contact exposure than feeding, possibly related to the suppression of both esterase and AChE activity caused by contact with eugenol. This difference suggests that the bee’s digestive tract may be able to process the compound more readily when it is fed than when it is absorbed through the cuticle. Alternately, eugenol may, in its vapor phase, be readily absorbed via the bee’s spiracles [75].

The five chemicals tested represent a broad range of categories of natural plant metabolites; as such, biochemical effects on honey bees varied widely. Among the various sublethal effects of the tested chemicals, transcripts of P450 were affected by four of the tested compounds: CYP-6A13 decreased when bees were exposed to bisabolene by contact but increased when bees were fed EcoTec+; CYP-6AQ1 was suppressed only by cineole and only on contact; and CYP-9Q1 was inhibited by feeding cinnamaldehyde and contact with cineole, but it increased when bees came into contact with bisabolene or were fed cineole. CYP-9Q2 also increased when bees were fed cineole. AChE activity was inhibited by contact with bisabolene and eugenol and by the ingestion of cinnamaldehyde. However, contact with cinnamaldehyde increased AChE activity. Cineole and EcoTec+ appeared to have no significant effect on AChE activity. We had expected more widespread suppression of AChE, given that EOs are hydrophobic and reportedly act on the nervous system as enzyme inhibitors [23,76] and specifically inhibit AChE, as Calva et al. [77] showed in EO from *Eugenia valvata*, which mostly consists of a-pinene, a monoterpene. In contrast, Rizvi et al. [78] measured the inhibition of AChE in Asian citrus psyllid by carvacrol and a-bisabolol, but chamazulene caused an increase in AChE activity. However, Calva et al. also posited that inhibition is likely due to an “undefined interaction” between AChE and the complex of EO components. Studies of measured mixtures of these compounds may shed light on unexpected metabolic responses. Esterase activity was suppressed by feeding bisabolene and cineole and by contact with eugenol. GST activity was increased by contact with bisabolene, though the correlation coefficient for this interaction was low, suggesting that other factors play a strong role in this change.

These widely varying effects of EO compounds on honey bee P450 transcripts exemplify the highly specific responses of the honey bee immune system to natural plant compounds, as previous research has shown. For example, Boncristiani et al. [79] saw the suppression of CYP-306A1 by thymol and hypothesized that this involved the inhibition of the protein kinase pathway related to the insect hormone 20-hydroxyecdysone. However, Gashout et al. [80] saw the suppression of CYP-9Q3 by thymol and showed evidence that thymol binds to octopamine and GABA receptors. Later, Ewert et al. [51] showed that thyme oil suppressed CYP-6AS1 but did not affect CYP-9A3, CYP-6AS14, or CYP-306A1. Mao et al. [81] showed that CYP-6AS genes are important in metabolizing flavonoids, particularly quercetin. Thus, one expects that cinnamaldehyde, also a flavonoid, would activate those genes as well.

While certain plant compounds were found to synergize with PBO to combat pyrethroid-resistant mosquitos in a previous study [82], no interaction was observed between the compounds tested and PBO, TPP, or DEM.

It may be noted that several statistically significant results show a low correlation, indicating low explanatory power in the independent variable. Because the hallmark of a healthy honey bee colony is high genetic variability [83], the individual worker response to an external stimulus is necessarily highly variable as well. This is particularly evident when the number of samples per treatment is low, such as in this study’s analysis of P450 transcripts (*n* = 3). In addition, the interaction between these natural compounds and honey bee metabolism is not fully understood and is likely complex.

Although individual consumption of sucrose solution was measured based on the number of surviving bees at the end of each feeding bioassay, the results were inconsistent and therefore not analyzed statistically. Sugar and water vials are positioned atop each cage, and dripping may produce erratic results. Consumption is an important parameter to reveal the stomach toxicity of chemicals. Improving feeding methods to prevent dripping is necessary in future studies.

These combined results align with previous studies on essential oils, indicating mainly sublethal effects of EOs on honey bees [84,85]. This supports the hope that these essential oils may pose less risk to non-target organisms compared to synthetic pesticides. However, the practical use of EOs is often limited by their low solubility and high volatility [86], challenges that were addressed in this study, and must be considered by end-users as well. For instance, the application instructions for EcoTec+ recommend constant agitation after dilution with water to avoid separation. Nanoemulsion through Emulsion Phase Inversion has been suggested as a method to overcome these challenges [9].

As EOs and their components are considered for widespread use to control various pests, an important consideration is the regulation of such chemicals as biopesticides. While the EPA registration process tends to be much quicker (typically one year versus three years for synthetic pesticides), the standards for their effects on non-target organisms are just as high [87]. This study provides insights into the impact of a selection of compounds that show promise as biopesticides. Ongoing studies are essential for both whole essential oils and active ingredients to ensure such chemicals truly represent a sustainable improvement in current synthetic pesticides. In addition, this study of the acute exposure of honey bees to EOs must be complemented by further research on chronic exposure to those EOs if we are to understand the real-world effects of these compounds.

## 5. Conclusions

In conclusion, the EOs examined in this study generally exhibited low toxicity to honey bees; cinnamaldehyde displayed toxicity when ingested, while eugenol was toxic on contact. The biochemical effects varied among compounds but generally remained sublethal, with some even showing potentially beneficial effects. Further research is necessary to comprehensively understand the long list of EO active components and their impacts on non-target organisms as they are integrated into pest control strategies.

## Figures and Tables

**Figure 1 insects-16-00303-f001:**
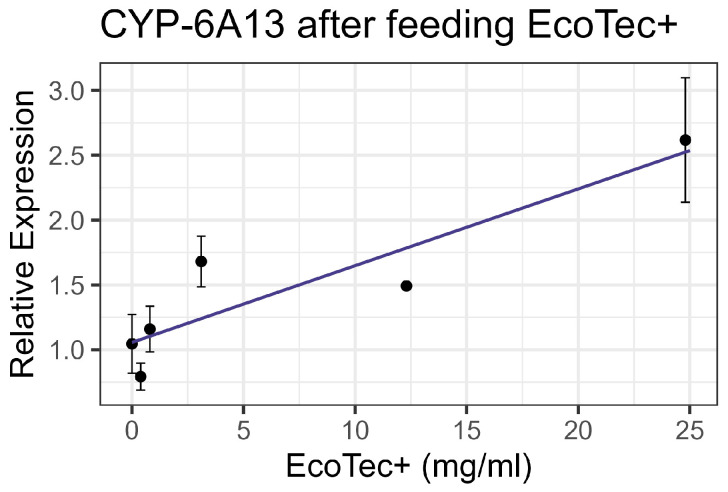
CYP-6A13 transcript levels after feeding with EcoTec+, fit to linear regression model.

**Figure 2 insects-16-00303-f002:**
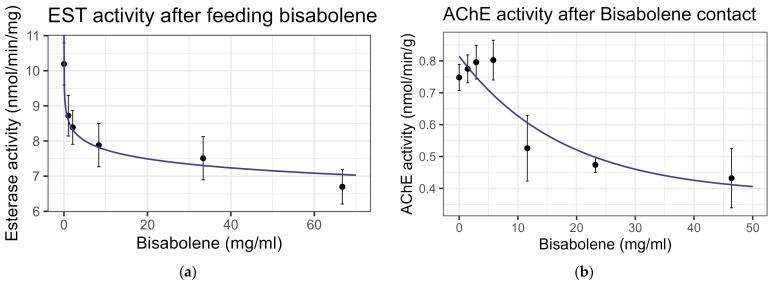
Enzyme activity after exposure to bisabolene: (**a**) esterase activity after feeding; log-linear trend is shown; (**b**) acetylcholine esterase activity after contact, fit to exponential decay model. Points and error bars represent mean ± standard error for each dose in this figure and all following figures.

**Figure 3 insects-16-00303-f003:**
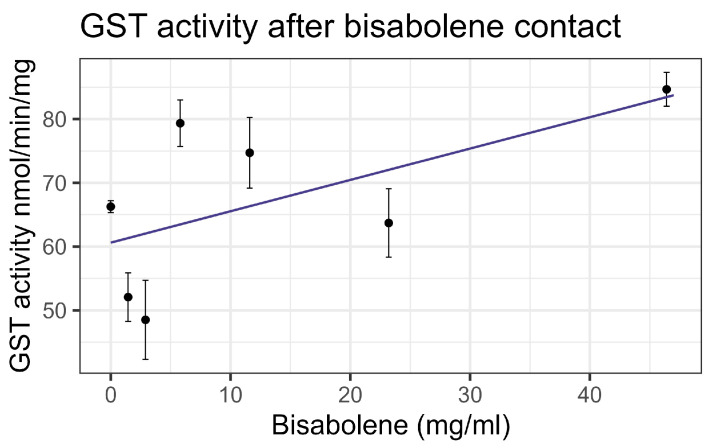
Glutathione-S-transferase activity affected by contact with bisabolene, fit to a linear regression model.

**Figure 4 insects-16-00303-f004:**
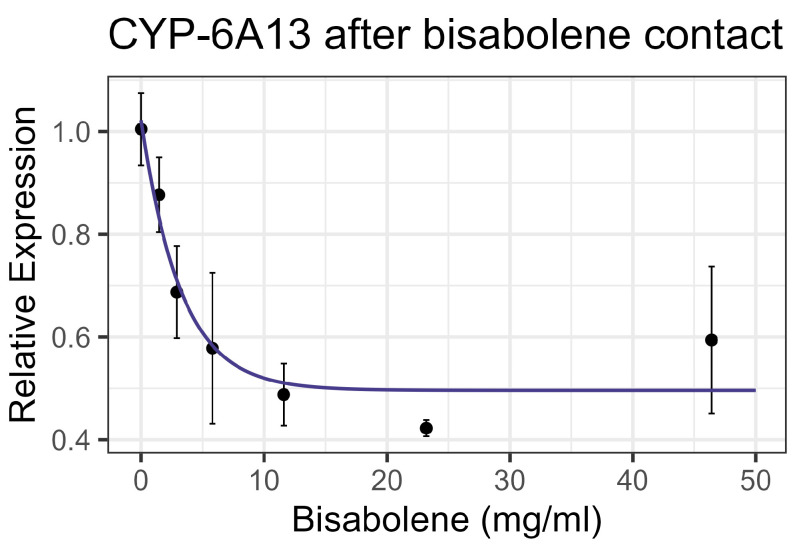
CYP-6A13 P450 transcript levels after contact with bisabolene, fit to an exponential decay model.

**Figure 5 insects-16-00303-f005:**
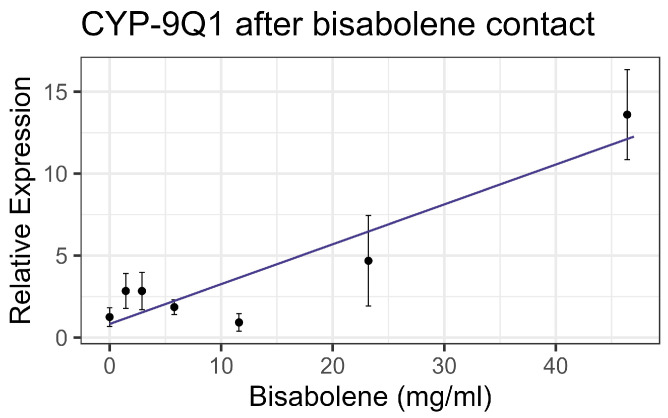
CYP-9Q1 P450 transcript levels after contact with bisabolene, fit to a linear regression model.

**Figure 6 insects-16-00303-f006:**
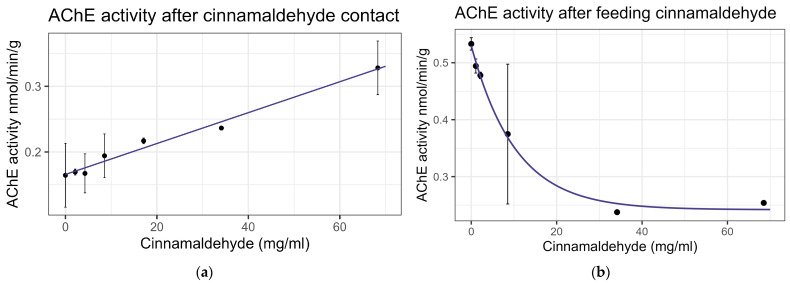
Acetylcholine esterase activity after exposure to cinnamaldehyde: (**a**) after contact; linear trend is shown; (**b**) after feeding, fit to exponential decay model.

**Figure 7 insects-16-00303-f007:**
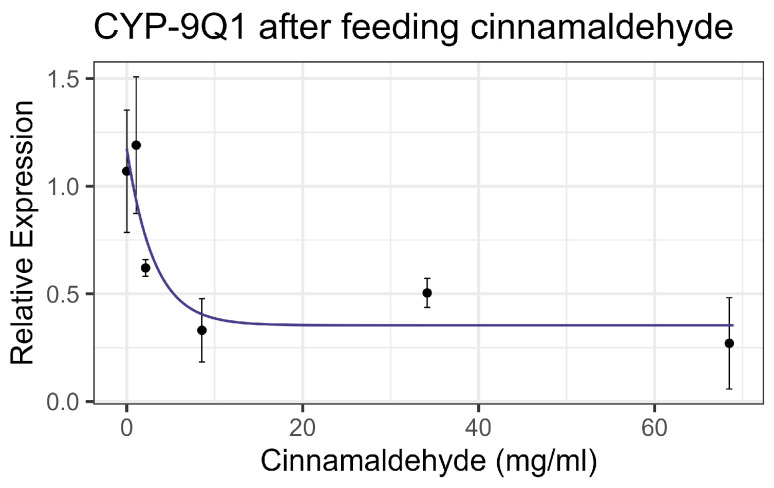
CYP-9Q1 transcript levels after feeding cinnamaldehyde, fit to an exponential decay model.

**Figure 8 insects-16-00303-f008:**
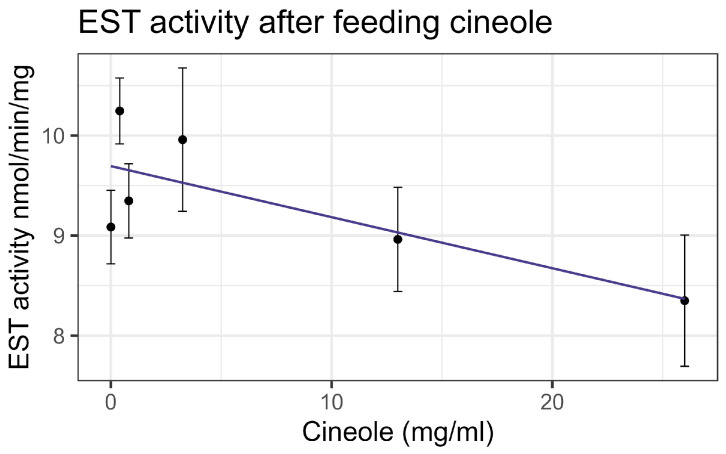
Esterase activity affected by feeding cineole, fit to a linear regression model.

**Figure 9 insects-16-00303-f009:**
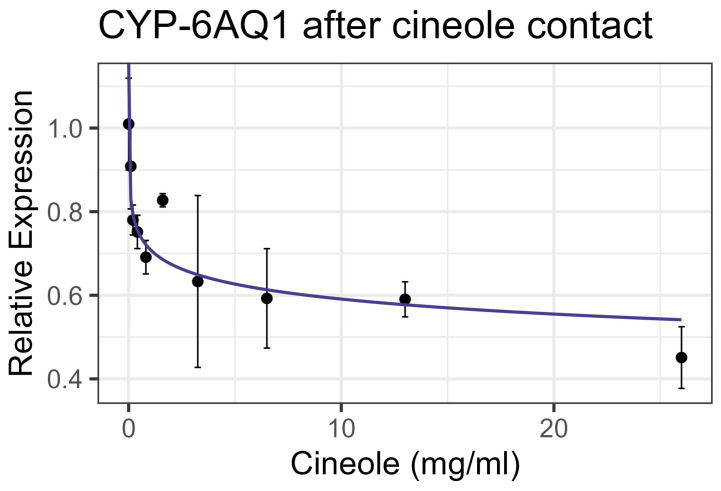
CYP-6AQ1 transcript levels after contact exposure to cineole, fit to an exponential decay model.

**Figure 10 insects-16-00303-f010:**
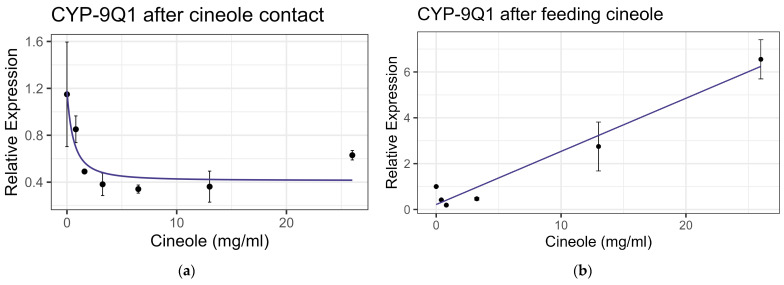
CYP-9Q1 P450 transcript levels after exposure to cineole: (**a**) after contact fit to an exponential decay model; (**b**) after feeding, fit to a linear regression model.

**Figure 11 insects-16-00303-f011:**
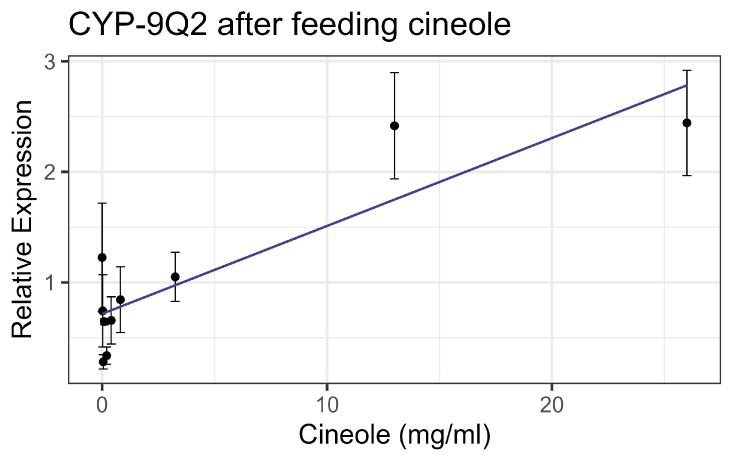
CYP-9Q2 transcript levels after feeding cineole, fit to a linear regression model.

**Figure 12 insects-16-00303-f012:**
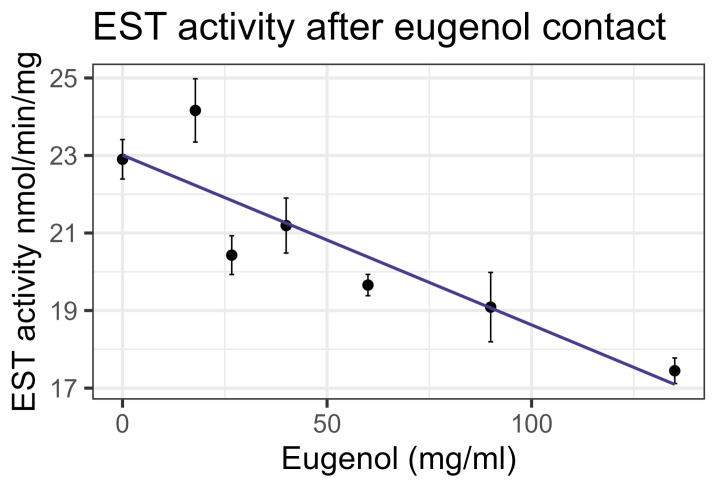
Esterase activity affected by contact with eugenol, fit to a linear regression model.

**Figure 13 insects-16-00303-f013:**
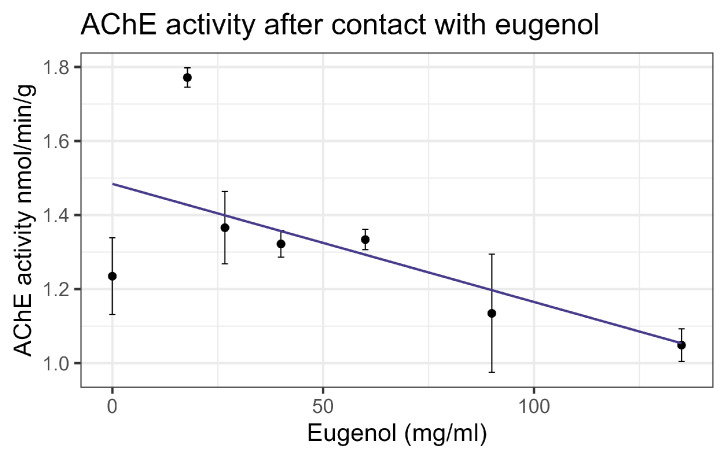
Acetylcholinesterase activity affected by contact with eugenol, fit to a linear regression model.

**Table 1 insects-16-00303-t001:** P450 primer sequences used for qRT-PCR.

Primer Name	Sequence
CYP-9Q1 (LOC408452) forward	CCACTCTGATAGGGTTGAAGAAG
CYP-9Q1 (LOC408452) reverse	TCGGTGAAGAACTTGGCTATG
CYP-9Q2 (LOC410492) forward	CAATTGAAGGAGCTGGGAAAG
CYP-9Q2 (LOC410492) reverse	CTCTATTGCGACAGCCTTTATTC
CYP-6AQ1 forward	AAACACACGTCAGCGGATAG
CYP-6AQ1 reverse	CCGAAGTAAAGACCGACGTAAG
CYP-6A13 (LOC727598) forward	CGGAACCTGAAGTATTCGATCC
CYP-6A13 (LOC727598) reverse	GATGCAATTTCTTGGCCCATC

**Table 2 insects-16-00303-t002:** Honey bee mortality after 48 h of being fed cinnamaldehyde.

Cinnamaldehyde, mg/mL	% Mortality, 48 h	SEM
0.0	6.0	2.4
1.1	18.0	4.4
2.1	33.8	2.4
8.5	45.0	2.2
34.2	76.0	14.8
68.5	89.0	6.2

**Table 3 insects-16-00303-t003:** Honey bee mortality 48 h after contact with eugenol.

Eugenol, mg/mL	% Mortality, 48 h	SEM
0.0	0.0	0.0
16.9	0.0	0.0
26.7	15.0	15.0
33.8	12.5	7.5
40.0	12.5	7.2
60.0	45.0	7.4
67.5	31.3	16.0
90.0	86.7	13.3
135.0	90.0	5.0

## Data Availability

The data that support the findings of this study are available upon request from the corresponding author [Y.Z.].

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
