# Peer review of "Risk Assessment of Effects of Essential Oils on Honey Bees (Apis mellifera L.)"

_insects, 2025, doi:10.3390/insects16030303_

Round 1

Reviewer 1 Report

Comments and Suggestions for Authors

The paper entitled "Risk assessment of Essential Oils towards honey bees (Apis mellifera L.)" reports results that investigate the acute toxicity of essential oils, used at biopesticides, on honey bees, through contact and feeding exposure. This study is interesting because little is known about the toxicity of essentials oils, their potential sublethal effects. The outcomes of the study are interesting and clear. I have some major comments and suggestions: 

Simple summary

L15: remove ‘caused’ or write ‘reduced the protective enzyme activity’

Abstract 

There is no mention of your interactions between pesticides and essential oils.

L35: I suggest removing ‘pollinators’ and ‘sustainable’ and replace by ‘enzymes’ and ‘toxicity’.

Introduction 

Overall, the introduction is well written, we understand the approach of this study considering the context. I suggest adding a sentence on the last paragraph on the interaction between pesticides and oils that the authors carried out, even if there are no real effects.

Materials and Methods 

The methods are well-described, allowing replication of the study.

L89-90: Are your colonies treated with oil for varroa and small hive beetle at the same time you collected frames from them for your experiment? Usually, there is a need to wait 4 weeks after a potential treatment to use the bees for toxicity tests.

L122: This is confusing. The authors are talking about four types of bioassays, but they described only two types below, contact toxicity with the spray tower and feeding toxicity with sucrose solution. They also did not precise the mixtures that you tried: interactions between pesticides and essential oils. Be clearer on this part.

L136-137: write ‘days’ instead of ‘d’.

L133: Did the authors measure the individual consumption in the feeding bioassay?

L138: Please describe briefly the protocol used for the enzyme assays. Just add the important steps than just only reference it.

L157: Did the authors add Abbott correction to the data base on their control before running the probit analysis?

Results 

The results are structured logically, with each essential oil analyzed separately.

The R² values in multiple instances are low (e.g., R² = 0.21 for CYP-9Q1 with cineole). While the trends are statistically significant, a low R² suggests weak explanatory power. There is a need to discuss this briefly, the implications of the weak correlations on biological significance.

L197: ‘Contact exposure to cinnamaldehyde did not resulted in significant mortality’, should be "did not result in".

Please add in the manuscript all honey bee mortality data for each essential oil (as tables) or add some of them which are not significant in a supplementary file.

It would be interesting to make some graphs showing the survival curves between each treatment even or some boxplots representing the percentage of mortality at the different time points.

Please add the mortality results of the mixtures in a supplementary file.

Discussion 

In the discussion, some sentences are overly long and complex. Breaking them into shorter, the authors will enhance readability with clearer statements. Moreover, the authors cited relevant references but could add some.

Take a look at this review and studies:

  • Gostin, I.N.; Popescu, I.E. Evaluation of the Essential Oils Used in the Production of Biopesticides: Assessing Their Toxicity toward Both Arthropod Target Species and Beneficial Pollinators. Agriculture 2024, 14, 81. https://doi.org/10.3390/agriculture14010081
  • Ariana, A.; Ebadi, R.; Tahmasebi, G. Laboratory Evaluation of Some Plant Essences to Control Varroa destructor(Acari: Varroidae).  Appl. Acarol. 200227, 319–327.
  • Ebert, T. A., Kevan, P. G., Bishop, B. L., Kevan, S. D., & Downer, R. A. (2007). Oral toxicity of essential oils and organic acids fed to honey bees (Apis mellifera). Journal of Apicultural Research46(4), 220–224. https://doi.org/10.1080/00218839.2007.11101398

It would be interesting to mention more regulatory considerations (e.g., how the EPA assesses EO-based pesticides?).

L255: The authors cite the composition of EcoTec+ but no study is mention examining the toxicity of rosemary oil, geraniol and peppermint oil on honey bees. Add some references.

Some findings in our results suggest significant changes in enzyme activity or transcript levels, but their biological implications are still unclear.

L262: the authors observed an increased in CYP-6A13 P450 transcripts. Please add a discussion on this. Does it suggest a metabolic response? Any other studies looking at P450 genes expression after exposure to essential oils?

L268: Could the authors add a comparison factor between their LC50 found for Bisabolene and the LC50 of da Silva et al.?  

L273-286: The presence or lack of AChE inhibition is interesting and deserves a stronger discussion. The authors can potentially reference studies on EO metabolism differences between insects?

L286: The authors sowed a decease in CYP-9Q1 transcript levels for cineole (result part L220). Please discuss this. Does this indicate a potential detoxification mechanism in honey bees? Is this decrease biologically relevant in a real-world exposure scenario?

L294: Lack of explanation of the difference mortality between contact and oral exposure for this oil. Hypothesis?

L302-303: please split the sentence in two. ‘… users as well. For instance, …’

Conclusion

EOs need to be further investigated for their toxicity to bees. The study is only part of the challenge of examining acute exposure, but what about chronic exposure? If beekeepers use EOs to control varroa mites or other pathogens, they will apply them over a long period of time.

Author Response

The paper entitled "Risk assessment of Essential Oils towards honey bees (Apis mellifera L.)" reports results that investigate the acute toxicity of essential oils, used at biopesticides, on honey bees, through contact and feeding exposure. This study is interesting because little is known about the toxicity of essentials oils, their potential sublethal effects. The outcomes of the study are interesting and clear. I have some major comments and suggestions: 

Simple summary

L15: remove ‘caused’ or write ‘reduced the protective enzyme activity’

Done

Abstract 

There is no mention of your interactions between pesticides and essential oils.

Added, L33

L35: I suggest removing ‘pollinators’ and ‘sustainable’ and replace by ‘enzymes’ and ‘toxicity’.

done

Introduction 

Overall, the introduction is well written, we understand the approach of this study considering the context. I suggest adding a sentence on the last paragraph on the interaction between pesticides and oils that the authors carried out, even if there are no real effects.

Done, L85

Materials and Methods 

The methods are well-described, allowing replication of the study.

Thank you

L89-90: Are your colonies treated with oil for varroa and small hive beetle at the same time you collected frames from them for your experiment? Usually, there is a need to wait 4 weeks after a potential treatment to use the bees for toxicity tests.

Added clarification, L96

L122: This is confusing. The authors are talking about four types of bioassays, but they described only two types below, contact toxicity with the spray tower and feeding toxicity with sucrose solution. They also did not precise the mixtures that you tried: interactions between pesticides and essential oils. Be clearer on this part.

Added description of inhibitor and pesticide mixture bioassays, L141

L136-137: write ‘days’ instead of ‘d’.

done

L133: Did the authors measure the individual consumption in the feeding bioassay?

Explained individual consumption was measured but inconsistent and not analyzed, L429

L138: Please describe briefly the protocol used for the enzyme assays. Just add the important steps than just only reference it.

Added description of protein extraction and assays, L157-177

L157: Did the authors add Abbott correction to the data base on their control before running the probit analysis?

Yes.  Clarification added, L194

Add this reference: Abbott, W. S. 1925. A method of computing the effectiveness of an insecticide. J. Econ. Entomol. 18: 265-267.

Results 

The results are structured logically, with each essential oil analyzed separately.

Thank you

The R² values in multiple instances are low (e.g., R² = 0.21 for CYP-9Q1 with cineole). While the trends are statistically significant, a low R² suggests weak explanatory power. There is a need to discuss this briefly, the implications of the weak correlations on biological significance.

Discussed in Discussion, L422-428

L197: ‘Contact exposure to cinnamaldehyde did not resulted in significant mortality’, should be "did not result in".

corrected

Please add in the manuscript all honey bee mortality data for each essential oil (as tables) or add some of them which are not significant in a supplementary file.

Supplemental table created and referenced, L213

It would be interesting to make some graphs showing the survival curves between each treatment even or some boxplots representing the percentage of mortality at the different time points.

Supplementary table containing bioassay mortality was created; the number of graphs required for this seemed high and would increase the length of the paper significantly without greatly increasing information provided

Please add the mortality results of the mixtures in a supplementary file.

Done; see above supplementary table

Discussion 

In the discussion, some sentences are overly long and complex. Breaking them into shorter, the authors will enhance readability with clearer statements.

Broke up multiple long sentences

Moreover, the authors cited relevant references but could add some.

Take a look at this review and studies:

  • Gostin, I.N.; Popescu, I.E. Evaluation of the Essential Oils Used in the Production of Biopesticides: Assessing Their Toxicity toward Both Arthropod Target Species and Beneficial Pollinators. Agriculture 2024, 14, 81. https://doi.org/10.3390/agriculture14010081
  • Ariana, A.; Ebadi, R.; Tahmasebi, G. Laboratory Evaluation of Some Plant Essences to Control Varroa destructor(Acari: Varroidae).  Appl. Acarol. 200227, 319–327.
  • Ebert, T. A., Kevan, P. G., Bishop, B. L., Kevan, S. D., & Downer, R. A. (2007). Oral toxicity of essential oils and organic acids fed to honey bees (Apis mellifera). Journal of Apicultural Research46(4), 220–224. https://doi.org/10.1080/00218839.2007.11101398

Added these refs where appropriate

It would be interesting to mention more regulatory considerations (e.g., how the EPA assesses EO-based pesticides?).

Added, L443

L255: The authors cite the composition of EcoTec+ but no study is mention examining the toxicity of rosemary oil, geraniol and peppermint oil on honey bees. Add some references.

Done; L327-336

Some findings in your results suggest significant changes in enzyme activity or transcript levels, but their biological implications are still unclear.

More information are added to L73, L262, and L499, with 3 new citations as below.

Biological activities of essential oils from Moroccan plants against the honey bee ectoparasitic mite, Varroa destructor H Alahyane, M Ouknin, H Aboussaid… - … Journal of Acarology, 2022 - Taylor & Francis

Botanical insecticide as simple extractives for pest control

WM Hikal, RS Baeshen, HAH Said-Al Ahl - Cogent Biology, 2017 - Taylor & Francis

… Eucalyptus essential oil significantly reduced the number of … Essential oils act by inhibiting
insect acetylcholinesterase (… Botanicals chemicals that cause insects to make oriented …

Save Cite Cited by 391 Related articles All 8 versions 

Rizvi, et al (2018), Younsi, et al (2016) in discussion of AChE

L262: the authors observed an increased in CYP-6A13 P450 transcripts. Please add a discussion on this. Does it suggest a metabolic response? Any other studies looking at P450 genes expression after exposure to essential oils?

Added discussion on previous studies of honey bee p450 responses to essential oils; L408-418

L268: Could the authors add a comparison factor between their LC50 found for Bisabolene and the LC50 of da Silva et al.?  

Done, and adjusted the assessment, with a possible explanation of the difference; L344

L273-286: The presence or lack of AChE inhibition is interesting and deserves a stronger discussion. The authors can potentially reference studies on EO metabolism differences between insects?

Expanded discussion of AChE response; L395-404

L286: The authors showed a decrease in CYP-9Q1 transcript levels for cineole (result part L220). Please discuss this. Does this indicate a potential detoxification mechanism in honey bees? Is this decrease biologically relevant in a real-world exposure scenario?

Added discussion of varied p450 responses; L372-375

L294: Lack of explanation of the difference mortality between contact and oral exposure for this oil. Hypothesis?

Added hypothesis; L385

L302-303: please split the sentence in two. ‘… users as well. For instance, …’

done

Conclusion

EOs need to be further investigated for their toxicity to bees. The study is only part of the challenge of examining acute exposure, but what about chronic exposure? If beekeepers use EOs to control varroa mites or other pathogens, they will apply them over a long period of time.

Added; L449

Reviewer 2 Report

Comments and Suggestions for Authors

I have been asked to read and revised the MS by Caren et al., 2025 entitled “Risk Assessment of Essential Oils toward honey bees (Apis  mellifera L.)”.

The aim of the study is to examine the effects of EcoTec+, a commercially available essential oil mixture, and four essential oil components (β-bisabolene, cinnamaldehyde, 1,8-cineole, and eugenol) on honey bee workers through the topical or oral route of administration. Particularly, the responses of esterase (EST), glutathione-S-transferase (GST), acetylcholine esterase (AChE), and P450, were assessed.

The study is interesting as it focuses on a multidisciplinary topic with high applicative outcomes. However, I find that there are a major points that should be dealt with before considering the MS for publication.

Mainly, in the MM section (ll 121-122) it is stated that “Four types of bioassays were conducted for each chemical: contact gradient, feeding gradient, inhibition (contact) and mixtures (contact)”, however it appears that the inhibition and mixtures test are not described in the MM section nor deeply analyzed in the Results and Conclusion sections. I would suggest to break the study in two and deepen the aspects of inhibition and mixture tests in a second MS.

The MM section should more clearly describe the experimental design as important details are missing. Please, see below some suggestions:

L 95: “Newly emerged worker bees from different colonies were pooled, without controlling for  the colony effect”. What do you mean by “colony effect”? please clarify.

L123: “Bees were sampled from each bioassay at given times after exposure”, please indicate here the exposure times.

L126: Please specify the concentrations.

L128: “500 μl of the pesticide/essential oil solution” does the bar stand for “or”? is it a ratio? Please clarify.

Also, why do Authors only show partially their results in the figures? I find that the Results section could be improved by reporting the figures of the activity of the different enzymes for all the analyzed EOs and the different routes of exposure (which should be indicated in the Figure. As an example, if you look at Figures 6 and 7 without reading their descriptions they give the impression of being the same thing). The results of the control should also be presented in order to make comparisons. Furthermore, why is mortality only presented for  Eugenol? It should be presented for each compound, at least as a total value. I believe Authors have invested a great amount of time and efforts in this study, however at the moment the Results section appears incomplete. Full results should be presented, if not in the main text as Supplementary material.

As previously said, there is a lot of data collected. In order for it to be fully presented I suggest to focus on the main aim of the study (the effects of single compounds) and present all data available in a thoroughly revised manner.

Author Response

I have been asked to read and revised the MS by Caren et al., 2025 entitled “Risk Assessment of Essential Oils toward honey bees (Apis  mellifera L.)”.

The aim of the study is to examine the effects of EcoTec+, a commercially available essential oil mixture, and four essential oil components (β-bisabolene, cinnamaldehyde, 1,8-cineole, and eugenol) on honey bee workers through the topical or oral route of administration. Particularly, the responses of esterase (EST), glutathione-S-transferase (GST), acetylcholine esterase (AChE), and P450, were assessed.

The study is interesting as it focuses on a multidisciplinary topic with high applicative outcomes. However, I find that there are major points that should be dealt with before considering the MS for publication.

Mainly, in the MM section (ll 121-122) it is stated that “Four types of bioassays were conducted for each chemical: contact gradient, feeding gradient, inhibition (contact) and mixtures (contact)”, however it appears that the inhibition and mixtures test are not described in the MM section nor deeply analyzed in the Results and Conclusion sections. I would suggest to break the study in two and deepen the aspects of inhibition and mixture tests in a second MS.

Added more complete description of inhibition and mixture bioassays in MM (L141-147). Noted elsewhere that results of these bioassays were not significant

The MM section should more clearly describe the experimental design as important details are missing. Please, see below some suggestions:

L 95: “Newly emerged worker bees from different colonies were pooled, without controlling for  the colony effect”. What do you mean by “colony effect”? please clarify.

This was required by supervisor, without explanation. Removed.

L123: “Bees were sampled from each bioassay at given times after exposure”, please indicate here the exposure times.

Clarified: 24h for contact and 7 days for feeding bioassays

L126: Please specify the concentrations.

Added standard concentrations; L132

L128: “500 μl of the pesticide/essential oil solution” does the bar stand for “or”? is it a ratio? Please clarify.

Replaced ‘/’ with ‘or’

Also, why do Authors only show partially their results in the figures? I find that the Results section could be improved by reporting the figures of the activity of the different enzymes for all the analyzed EOs and the different routes of exposure

There would be a total of 70 graphs if we were to include all non-significant data plots; this would greatly increase the total length of the paper without adding real content. We have included supplemental materials with all results, however.

(which should be indicated in the Figure. As an example, if you look at Figures 6 and 7 without reading their descriptions they give the impression of being the same thing).

Added chemical and exposure type to graph titles

 The results of the control should also be presented in order to make comparisons. Furthermore, why is mortality only presented for  Eugenol? It should be presented for each compound, at least as a total value. I believe Authors have invested a great amount of time and efforts in this study, however at the moment the Results section appears incomplete. Full results should be presented, if not in the main text as Supplementary material.

Supplementary table containing bioassay mortality was set up

As previously said, there is a lot of data collected. In order for it to be fully presented I suggest to focus on the main aim of the study (the effects of single compounds) and present all data available in a thoroughly revised manner.

Two supplementary tables were added, and Discussions were sub

Round 2

Reviewer 1 Report

Comments and Suggestions for Authors

I appreciated the authors' effort to improve the manuscript, taking into account the feedback from reviewers. 

I had nothing to add. 

Reviewer 2 Report

Comments and Suggestions for Authors

I would like to thank very much the Authors for taking into consideration my suggestions and spending time in revising the MS, I hope you can appreciate the improvements.

 All the missing data in the MM and Results Sections are now presented, and all points were clarified.  I would just present the Supplementary tables in separate files as requested by the Journal.

This said, I believe that the MS is now suitable for publication.